# A Pentaband Compound Reconfigurable Antenna for 5G and Multi-Standard Sub-6GHz Wireless Applications

**Ikhlas Ahmad [1], Wasi Ur Rehman Khan [1], Haris Dildar [1], Sadiq Ullah [1,\*], Shakir Ullah [1], Naveed Mufti [1], Babar Kamal [2], Toufeeq Ahmad [1], Adnan Ghaffar [3] and Mousa I. Hussien [4,\*]**

1   Department of Telecommunication Engineering, University of Engineering & Technology, Mardan 23200, Pakistan; ikhlasahmed725@gmail.com (I.A.); wasi.khan@uetmardan.edu.pk (W.U.R.K.); haris8371@gmail.com (H.D.); shakirhayat.eng@gmail.com (S.U.); naveed@uetmardan.edu.pk (N.M.); drtoufeeq@uetmardan.edu.pk (T.A.)
2   Center of Intelligent Acoustics and Immersive Communications, Northwestern Polytechnical University, Xi'an 710072, China; babarkamal.55@mail.nwpu.edu.cn
3   Department of Electrical and Electronic Engineering, Auckland University of Technology, Auckland 1010, New Zealand; aghaffar@aut.ac.nz
4   Department of Electrical Engineering, United Arab Emirates University, Al Ain 15551, United Arab Emirates
\*   Correspondence: sadiqullah@uetmardan.edu.pk (S.U.); mihussien@uaeu.ac.ae (M.I.H.)

**Abstract:** This work proposes a low-profile, printed antenna that offers pattern and frequency reconfiguration functionalities printed on FR-4 substrate with a size of $46 \times 32 \times 1.6 \text{ mm}^3$. The proposed antenna can operate in five different frequency bands, each one identified as a Mode, wherein there are possibilities of pattern reconfiguration. The frequency and pattern reconfigurability are made possible through 12 p-i-n diode switches (S1 to S12). The former is enabled through the switches S1 to S4 within the radiating patch, hence effectively controlling the resonant bands of the antenna; the latter is made possible through main lobe beam steering, enabled by the rest of the eight switches (S5 to S12), loaded in split parasitic elements designed on both sides of the radiator. The proposed antenna operates in the 5 GHz (4.52–5.39 GHz) band when all switches are OFF. When S1 is ON, the operating band shifts to 3.5 GHz (2.96–4.17 GHz); it changes to a 2.6 GHz (2.36–2.95 GHz) band when S1 and S2 are ON. When S3 is also turned ON, the antenna shifts to the 2.1 GHz Band (1.95–2.30 GHz). When S1–S4 are ON, the operating band shifts to a 1.8GHz (1.67–1.90 GHz) band. In all these bands, the return loss remains less than −10 dB while maintaining good impedance matching. At each operating band, the ON/OFF states of the eight p-i-n diode switches (S5 through S12) enable beam steering. The proposed antenna can direct the main beam in five distinct directions at 3.5GHz, 2.6 GHz, and 2.1 GHz bands, and three different directions at 5 GHz and 1.8 GHz bands. Different 5G bands (2.1, 2.6, 3.5, and 5) GHz, which fall in the sub 6GHz range, are supported by the proposed antenna. In addition, GSM (1.8 GHz), UMTS (2.1 GHz), 4G-LTE (2.1 GHz and 2.6 GHz), WiMAX (2.6 GHz and 3.5 GHz) and WLAN (5 GHz) applications are also supported by the proposed antenna, which is a candidate for handheld 5G/4G/3G devices.

**Keywords:** low-profile; compound reconfigurable antenna; 5G; beam steering

## 1. Introduction

As modern wireless communication standards and technologies continue to evolve, and multi-band, multi-user requirements emerge, the wireless access networks of 5G and future generation mobile networks are expected to support several connections simultaneously and are low profile. The term "low profile" means the overall height/thickness of the antenna is less than λ/10. The Federal Communications Commission (FCC) has identified three broad frequency ranges for 5G-mm-Wave, sub-6 GHz and sub-1 GHz, designated as high (>24 GHz band and higher), medium (1–6 GHz), and low (<1 GHz) bands, respectively [1,2]. The low band offers deep and widespread 5G coverage, the mm-Wave offers

super-fast data rates and large channel capacity, and a combination of both is offered by the medium band. So, 5G coverage and speed are determined by the band in use. Very high data rates can be achieved by using the high band, although there are challenges in the implementation of mm-Wave communication with mobility. Atmospheric attenuation and other factors do not allow long range propagation of mm-wave range frequencies. The low band spectrum offers blanket coverage but at the price of the data rate. Conversely, sub-6 GHz frequencies offer comparatively longer propagation ranges and reasonably high data rate communication, hence its potential application in both urban and rural areas for 5G network implementation. While 5G is implemented globally, the network and operations previously working below 6 GHz are switching to 5G, even while the mm-wave component of 5G is not as mature as sub-6 GHz. Due cognizance also must be given to the fact that multiple mobile generation networks operate and co-exist in sub-6 GHz bands. With the advancement in wireless communication technology and the requirements of multi-band operation, reconfigurable antennas have demand and relevance in cellular mobile communication [3]. Frequency reconfigurable antennas provide efficient utilization of the spectrum by tuning the operating frequency to the intended frequency bands [4]. On the other hand, pattern reconfigurable antennas offer enhanced gain, energy saving [5], reduced effect of co-channel interference and improved channel capacity [6,7], by redirecting the main radiation lobe towards the intended direction. The combination of both these features, as reported in this work, can offer enhanced features and capabilities of relevance to 5G, especially in sub-6 GHz bands. Most of the previously published work reports the use of one of these reconfiguration techniques. In [8,9], the review about reconfigurable antennas is presented, reporting recent developments in reconfigurable antennas.

In [10], a low-profile MIMO array antenna for millimeter wave 5G application is reported. The antenna covers 28.2 to 30.7 GHz frequencies. A simple structure with an arrow-shaped radiator and matching stub frequency reconfigurable antenna to cover a 5G sub-6 GHz band is presented in [11]. Reference [12] proposes a unique-shaped antenna for multiple applications (UMTS, WLAN and Wi-MAX) and offers frequency reconfigurability. In [13], a slotted structured antenna that is frequency reconfigurable is introduced. The antenna has two slots in the main radiator and a single ring slot on the ground plane, loaded with two pin diodes switches. Similarly, another frequency reconfigurable antenna with a slotted structure in the radiator is presented in [14]. Two pin diodes have been used for a switching mechanism, inserted in a V-shaped slot in the radiator. In [15], a monopole frequency reconfigurable antenna for LTE applications is reported. The antenna is loaded with a reconfigurable composite right/left-handed (CRLH) unit cell. Two varactor diodes are used as actuators resulting in covering various LTE bands. A flexible frequency reconfigurable antenna for heterogenous applications is presented in [16]. The radiating patch is loaded with two pin diodes for altering the resonant length and providing various multi-band operations. In [17], a compact multi-mode frequency reconfigurable antenna for portable devices is introduced. The antenna is coplanar waveguide (CPW) fed, has an inverted triangle shaped main radiator and four parasitic patches loaded with three pin diodes switches. The antenna provides a wide band and four other resonant bands. For the Airport Surveillance Radar Band, Wi-Fi, WLAN, and ISM applications, [18] proposes a frequency reconfigurable, low-profile antenna. References [19,20] present a frequency reconfigurable antenna for 5G and WLAN applications. In [21], a dual-polarized, dual-band filtenna is proposed for 5G Base Station application, whereas [22] and [23] propose multi-band antenna operating in four and six bands, respectively, and offering frequency reconfiguration feature.

In [24], a Yagi-Uda antenna is proposed; it offers both omnidirectional and directional radiation pattern characteristics. Another pattern reconfigurable antenna for 1.8 GHz, capable of steering the beam in three distinct directions (78 degrees apart) is proposed by [25]. It also offers enhanced gain, made possible via a partially reflective surface (PRS). A pattern reconfigurable antenna, with a defected ground structure and slot tuning is proposed in [26]. In [27], an antenna with beam steering capability for a 2.45 GHz

band (wireless sensor network applications) is presented. Reference [28] reports a pattern reconfigurable antenna, whose unique feature is being single element, with large size. In [29], an antenna with a beam scanning range of −34° to +32° at 2.29 GHz is proposed for multiple autonomous applications. In [30], pattern reconfigurability is achieved by using a complementary split-ring resonators (CSRR) technique on the ground component and switches with dielectric resonators. In [31], micro-electro-mechanical switches (MEMS) are used with coupling cells to deliver similar results.

Despite the research discussed above, the need of the time is having both the reconfiguration features in a single antenna. In the published literature, there are a few studies that have proposed antennas offering both pattern and frequency reconfigurability features. Reference [32] proposes a CPW feed-based antenna, utilizing two pin-diodes for achieving frequency reconfigurability for 1.65–2.51 GHz bands, while pattern reconfigurability along +90°, +90° for the respective band. Two Varactor Diodes are used in a two-element array antenna in [33] for tuning to 2.38 GHz and 2.15 GHz and offering beam steering within −23° to +23°. In [34], another antenna that offers pattern and frequency reconfiguration is presented. It can operate in a smaller number of two frequency bands (28 and 38 GHz) and changes the beam after each 45° for eight patterns in different direction, by using a greater number of eighteen switches. Reference [35] presents frequency and pattern reconfigurable antenna, using pin diodes and Electromagnetic Bandgap (EBG) unit cells. The antenna is optimally suited for 2.4 and 5.8 GHz WLAN operation. In [36], a simple planar frequency and radiation pattern reconfigurable antenna was explained. One switch is inserted between different patches on the front side to resonate the different frequencies, and two diodes with an L-shaped stub were introduced on the back side with a partial ground plane used to switch between different radiation beams. Reference [37] presents an eight pin-diode switch-based pattern and frequency reconfigurable antenna, switching between 2.4 and 1.9 GHz bands, with beam shifting possible in two directions for each band. Reference [38] proposes a four pin-diode based, wideband slot antenna, offering operation in 3.6 GHz (3.4–3.8 GHz) and 4 GHz (3.7–4.2 GHz) Bands, with beam steering options at 20° and 25°. The most considerate design in [39] with small size and large bandwidth also imposes a challenge to use eight switches and the provision of 3.8 dB gain as a maximum.

This paper presents a compact sized (46 × 32 × 1.6 mm$^3$) composite pattern and frequency reconfigurable antenna. The combination reconfigurability is achieved via twelve pin-diodes switches, of which four switches (S1 and S4) enable frequency reconfigurability, while the remaining eight switches (S5–S12) allow beam steering. The antenna can operate within five different frequency bands, or simply, in five modes. Each mode further offers multiple beam steering cases over the operating frequency range. In Mode 1, the antenna operates at 5 GHz Band (4.52–5.39 GHz), while it resonates at 3.5, 2.6, 2.1 and 1.8 GHz bands in Modes 2, 3, 4 and 5, respectively. As showcased by simulation results, the proposed antenna is capable of beam steering in five different directions at 3.5 GHz, 2.6 GHz, and 2.1 GHz bands, and three distinct directions for 1.8 GHz and 5 GHz bands, with acceptable peak gain in every case. Several sub-6 GHz 5G bands (5, 3.5 GHz), as well as GSM Band (1.8 GHz), UMTS Band (2.1 GHz), 4G-LTE Band (2.1 GHz and 2.6 GHz), WiMAX Bands (2.6 GHz and 3.5 GHz), and WLAN Band (5 GHz) are supported by the proposed antenna. The main application for the proposed antenna is within 5G handsets, which are backward compatible with and support 4G and 3G services and offer WLAN connectivity.

A comparative analysis of the antenna proposed in the current work with other relevant previously published works demonstrates a superior performance of the proposed hybrid reconfigurable antenna, in terms of the number of operating bands and beams, as well as size and bandwidth.

This organization of this paper is as follows: the Design Methodology and Geometry of the proposed antenna are discussed in Section 2. Section 3 includes Experimental Results and Analysis. The final Section (Section 4) draws the conclusion of the current study.

## 2. Design Methodology and Geometry of the Proposed Antenna

Figure 1 shows the geometrical dimensions of the proposed hybrid antenna, whose parasitic elements, radiator, and ground have been printed on FR-4 substrate. The relative permittivity, $\varepsilon_r$, of the substrate is 4.3, while its loss tangent value, δ, is 0.025. The overall size of the proposed antenna is $46 \times 32 \times 1.6$ mm$^3$. A 3-mm wide, 50 Ω Microstrip line serves as feed. Table 1 specifies all the dimensions of the proposed antenna structure.

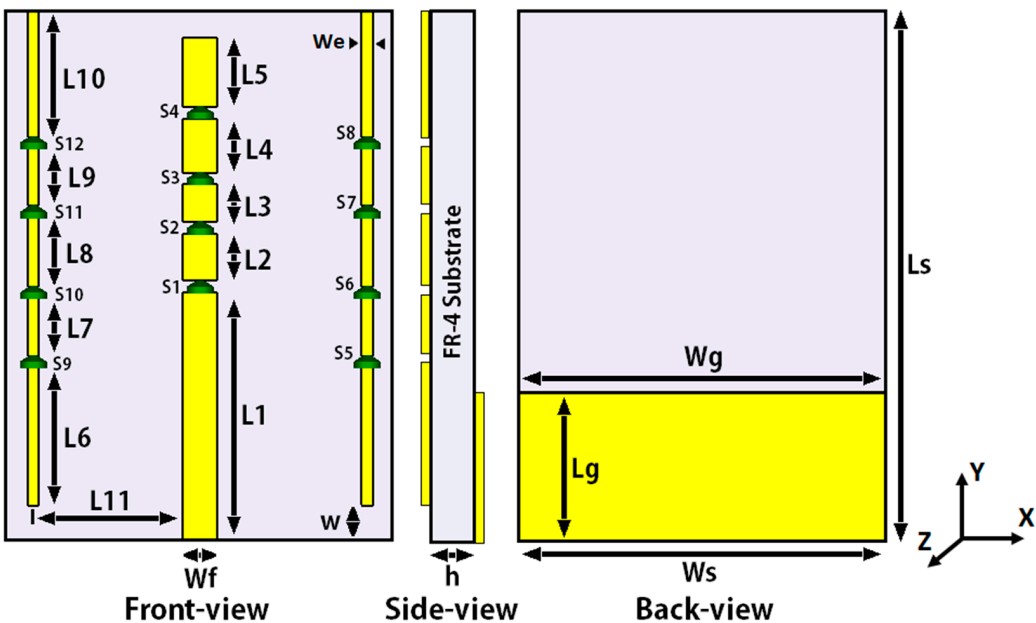

**Figure 1.** Geometric views of Hybrid antenna.

**Table 1.** Design Parameters of Hybrid Antenna.

| Parameter | Value (mm) | Parameter | Value (mm) |
|---|---|---|---|
| Ws | 32 | L4 | 4.7 |
| Ls | 46 | L5 | 6 |
| Wg | 20 | L6 | 12 |
| Lg | 12.5 | L7 | 5 |
| W | 3 | L8 | 6 |
| Wf | 3 | L9 | 5 |
| L1 | 21.5 | L10 | 11 |
| L2 | 4.1 | L11 | 12.5 |
| L3 | 3.3 | H | 1.6 |
| We | 1 | | |

For intended operating frequency $f$, the effective resonant length $L_f$ and effective permittivity $\varepsilon_{eff}$ are determined by using Equations (1) and (2).

$$L_f = \frac{c}{4f\sqrt{\varepsilon_{eff}}} \tag{1}$$

$$\varepsilon_{eff} = \frac{\varepsilon_r + 1}{2} + \frac{\varepsilon_r - 1}{2}\left(1 + 12\left(\frac{w}{h}\right)\right)^{-0.5}, \tag{2}$$

where $c$ represents the velocity of light in vacuum, $\varepsilon_r$ is the relative permittivity, $h$ is the height or thickness, and $w$ is the width of the substrate.

The radiating structure is designed near the center of the whole structure; the main radiator is complemented with four parasitic radiator patches, connected through pin diodes S1–S4 (Model SMP1345-079LF). The connections between the main radiator/patches are enabled/disabled by the ON, OFF states of pin diodes, acting as switches. This

effectively results in connecting or disconnecting patches from each other, and hence changing the operating frequency of the antenna, by varying its effective resonant length.

A 1 mm wide split parasitic elements of different lengths are introduced on both sides of the main radiator, as shown in Figure 1. Eight pin-diode switches are used between the parasitic elements, for pattern reconfiguration. The ON, OFF states of these pin diodes result in varying the contributing length of parasitic elements, resulting in varying actions of parasitic elements (reflector, director, or combination of both). These parasitic elements provide reasonable gain, impedance matching for all cases (C) of operating modes (M), in addition to beam steering capability.

### 2.1. Switching Technique

For any frequency band, the pin diodes (SMP1345-079LF) work as a potentiometer to vary the resistance. However, the working principle is different as the control element is the resonant length, which is varied to achieve frequency and pattern reconfigurability. Figure 2 shows the equivalent circuits when pin diodes are used in ON and OFF states. A simple RL series circuit is formed for the ON state with an extremely low resistor $R_L$ and an inductor $L$. For the OFF state, the circuit becomes RLC with a parallel inductor $L$, along with a resistor of a high value $R_h$ and a capacitor $C$. For fabrication, the model of p-i-n diode Skyworks SMP1345-079LF is chosen for its low cost and ease of availability. The parametric values from the datasheet can easily be modeled in the CST simulation environment as $C = 0.15$ pF, $L = 0.7$ nH and $R_L = 1.5$ $\Omega$.

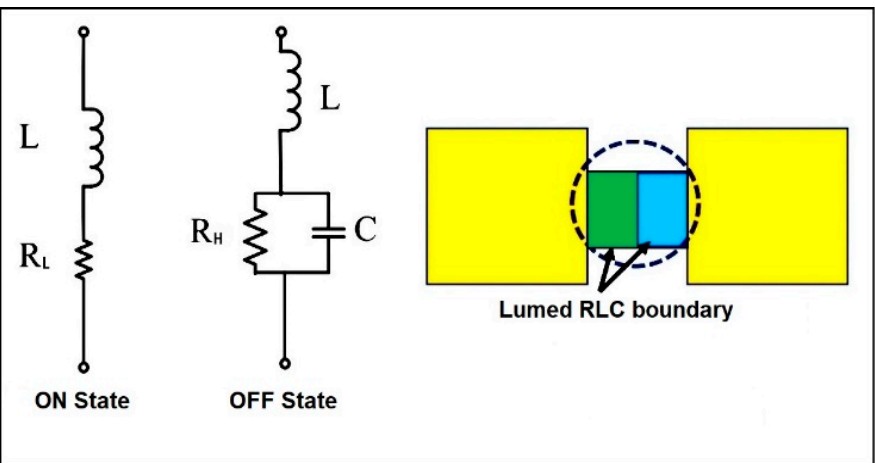

**Figure 2.** The equivalent circuits of pin diode and its CST model.

In the measurement step, the biasing circuit is placed on the back side of antenna for operating the pin diode in real time, as depicted in Figure 3.

### 2.2. Switching Configurations:

Table 2 enlists an overall feature set of 21 combinations, based on switching configurations for all operating modes (Modes 1–5) and their respective cases (C1 up to C5).

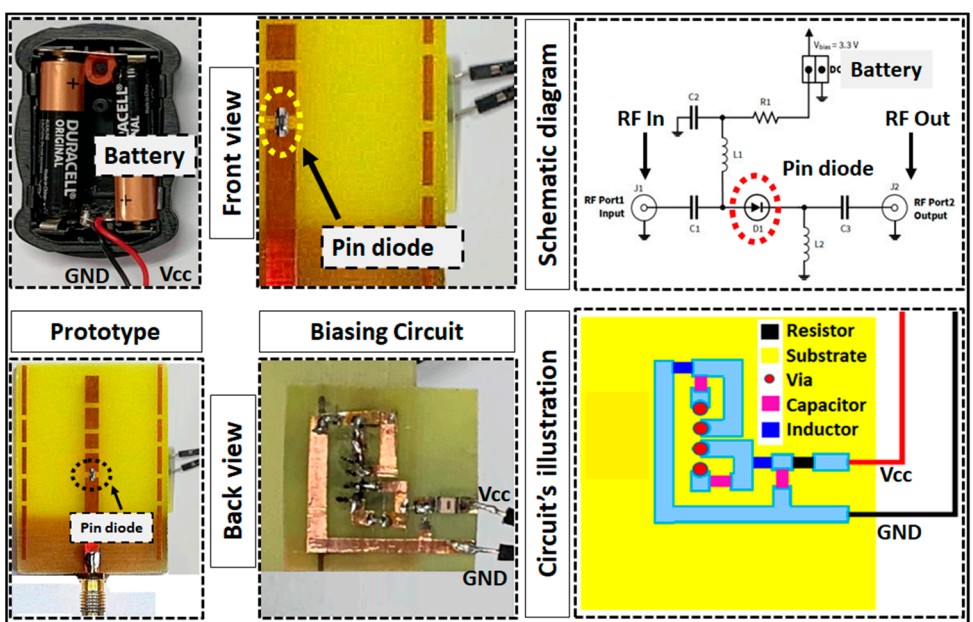

**Figure 3.** Biasing configuration of Mode 2 with fabricated prototype.

**Table 2.** Different mode of switching for frequency reconfiguration.

| Modes | Special Cases |
|---|---|
| **Mode-1** (ON: None, OFF: S1, S2, S3 & S4) | **C1** (OFF: S5–S12), **C2** (ON: S6 & S7), **C3** (ON: S10 & S11) |
| **Mode-2** (ON: S1, OFF: S2, S3 & S4) | **C1** (OFF: S5–S12), **C2** (ON: S6 & S7), **C3** (ON: S10 & S11), **C4** (ON: S5–S8, S11 & S12), **C5** (ON: S7–S12) |
| **Mode-3** (ON: S1 & S2, OFF: S3 & S4) | **C1** (OFF: S5-S12), **C2** (ON: S6, S7 & S8), **C3** (ON: S10, S11 & S12), **C4** (ON: S5–S8 & S10-S12), **C5** (ON: S6–S12) |
| **Mode-4** (ON: S1, S2, S3, OFF: S4) | **C1** (OFF: S5–S12), **C2** (ON: S6, S7 & S8), **C3** (ON: S10, S11 & S12), **C4** (ON: S5–S8 & S10–S12), **C5** (ON: S6–S12) |
| **Mode-5** (ON: S1–S4, OFF: None) | **C1** (OFF: S5–S12), **C2** (ON: S6, S7 & S8), **C3** (ON: S10, S11 & S12) |

### 2.3. Parametric Analysis:

To study the effect of variations of various parameters on the antenna's performance, a parametric study is conducted. The examination is done by considering parameters $L_{11}$ and $W_e$. The effect of the variation in $L_{11}$ and $W_e$ on $|S11|$ of the proposed antenna is shown in Figures 4 and 5, respectively. It has been observed from Figure 4 that, when $L_{11}$ is varied from 8 to 12.5 mm, the resonant bands are shifted towards the left side of the frequency axis. It is evident from Figure 5 that, if $W_e$ is doubled (i.e., 2 mm), then the impedance matching of the resonant bands is degraded.

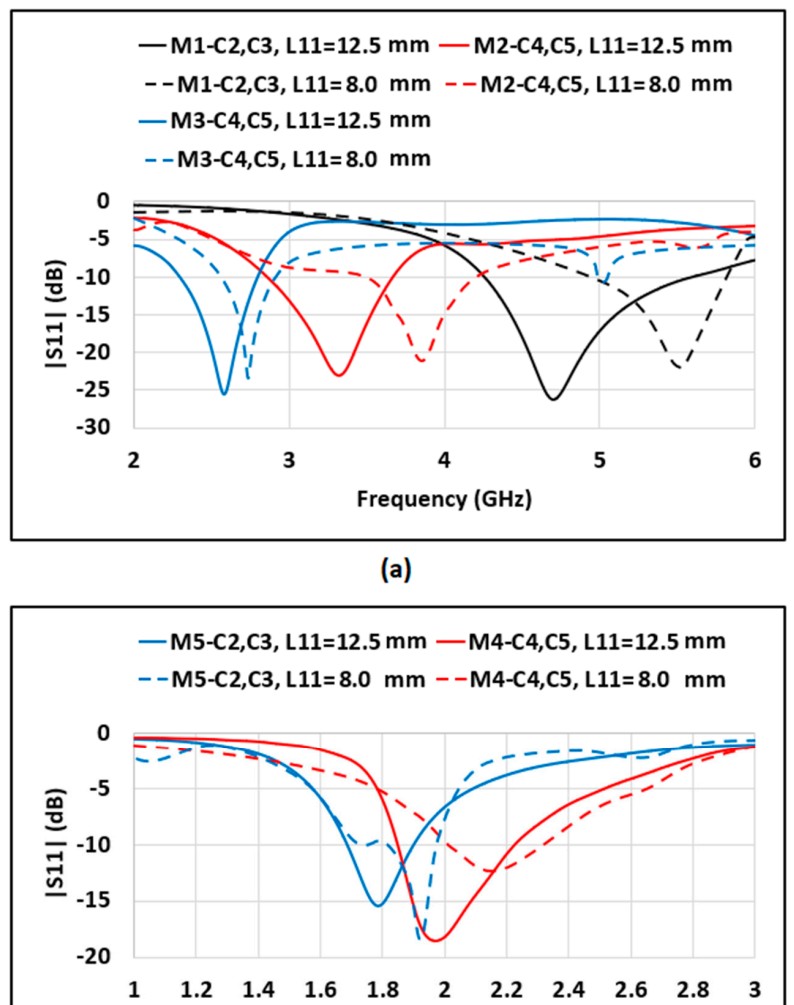

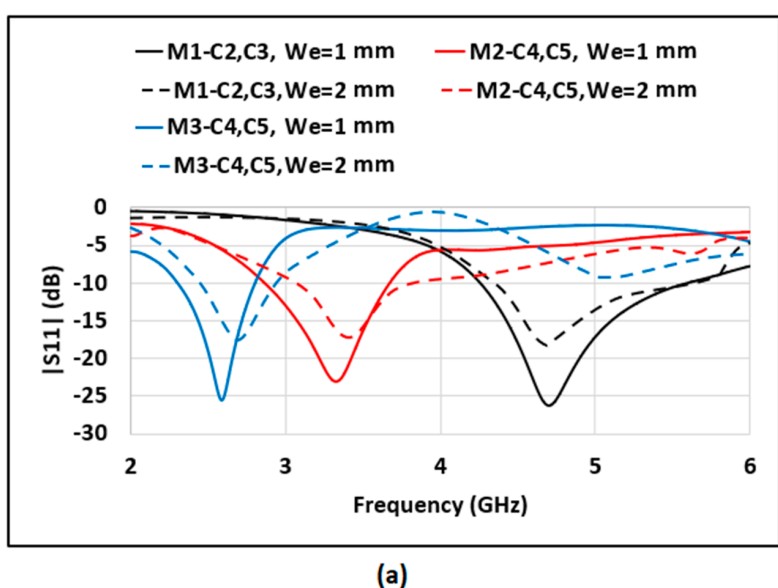

**Figure 4.** Effect of L11's variation on |S11| of various cases of operating modes (**a**,**b**).

**Figure 5.** *Cont.*

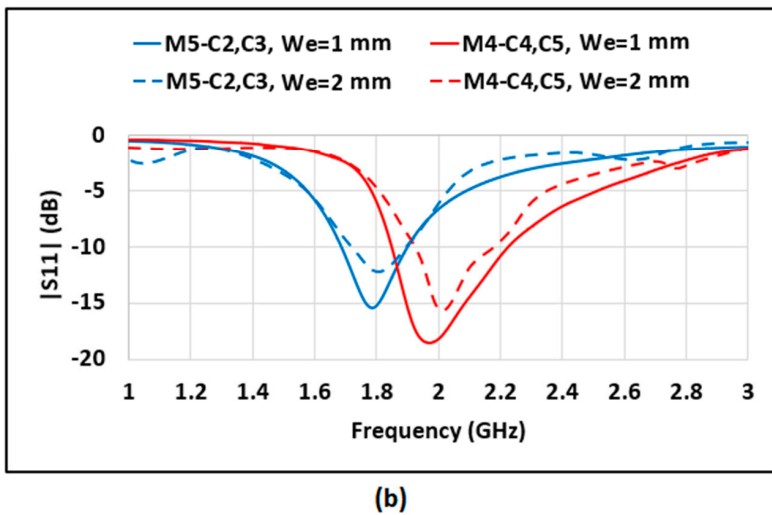

**(b)**

**Figure 5.** Effect of We's variation on |S11| of various cases of operating modes (**a**,**b**).

### 3. Experimental Results and Analysis

CST Microwave Studio has been utilized to design and simulate the proposed antenna structure, as well as to analyze its performance. Frequency reconfiguration is made possible through switches S1–S4, allowing five different modes of operation of the proposed antenna. Each mode enables the antenna to operate in different frequency band. Within each mode, for a certain frequency band, there are several cases for pattern reconfiguration, enabled by switches S5–S12. Figure 6 shows the photographs and the measurement setup of the fabricated prototype. Figures 7–9 show the return loss (S11) curves for all the five modes and their sub-cases. The gain vs frequency plots for the five switching modes are illustrated in Figure 10. Figure 11 shows the radiation patterns for the five modes, while Figure 12 displays the surface current distributions for all switching modes.

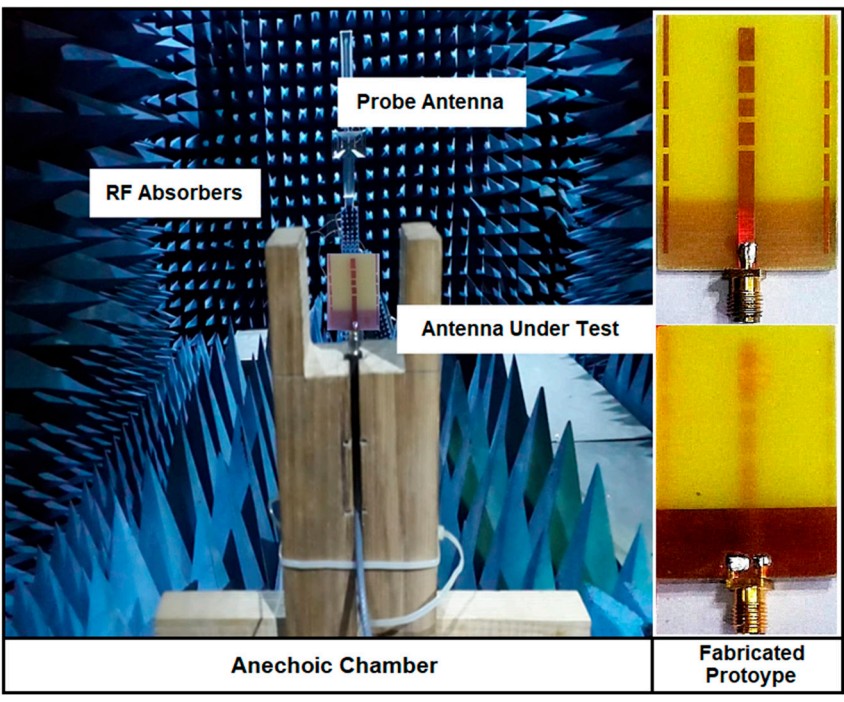

**Figure 6.** Experimental setup for measurement of radiation patterns.

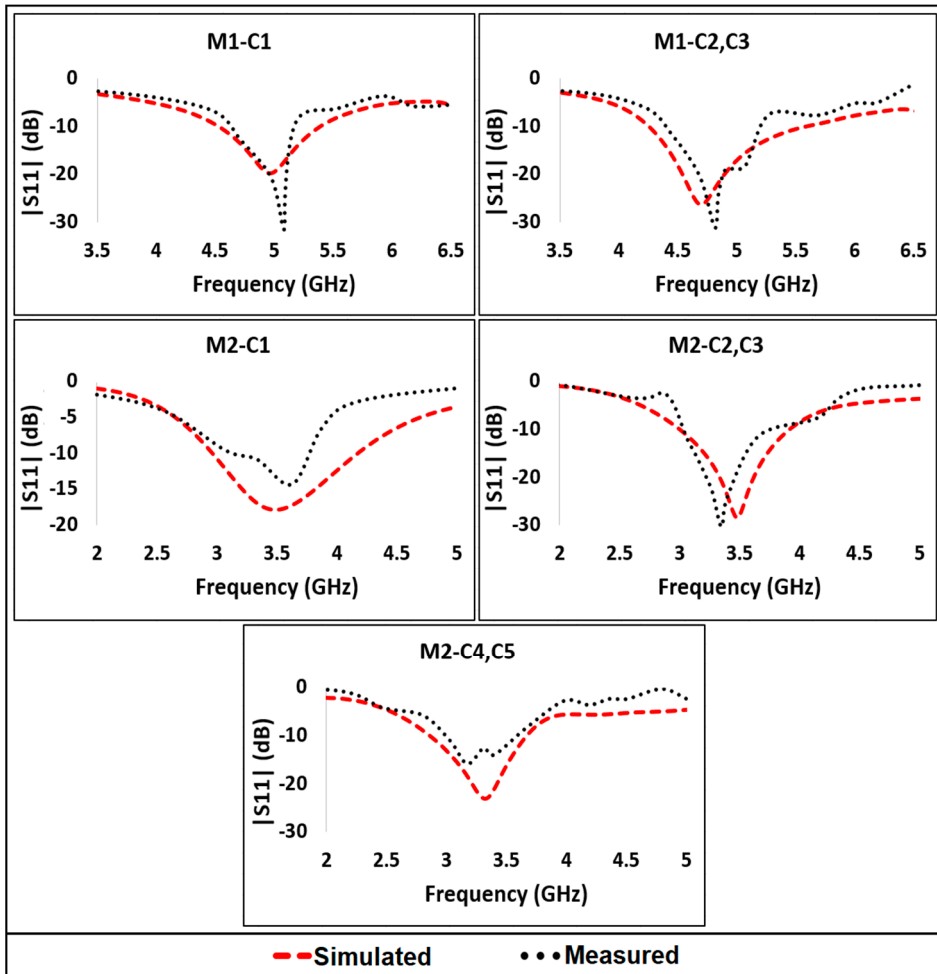

**Figure 7.** Measured and simulated |S11| for Cases of Modes 1 and Mode 2.

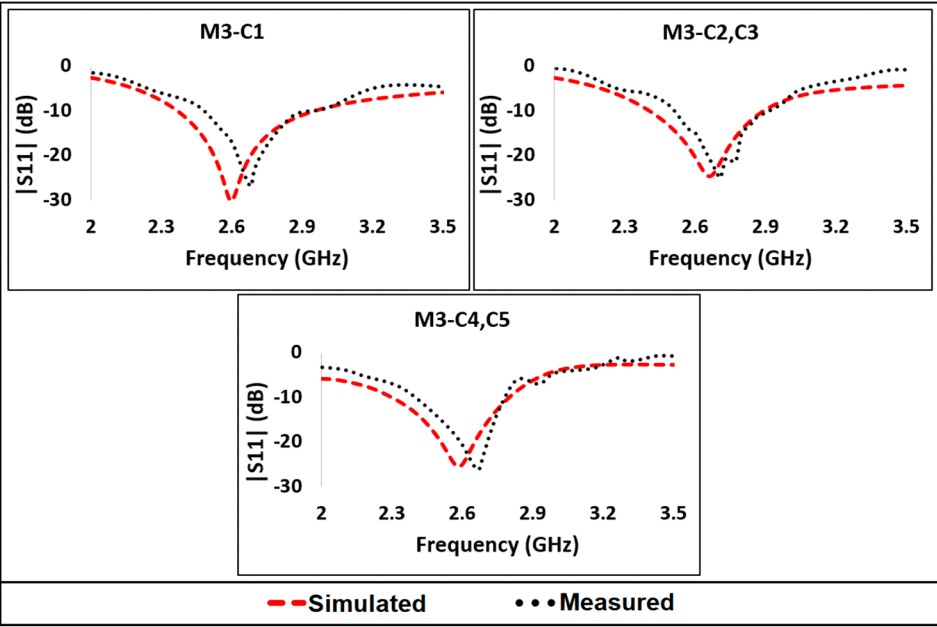

**Figure 8.** Measured vs. Simulated |S11| for Cases of Mode 3.

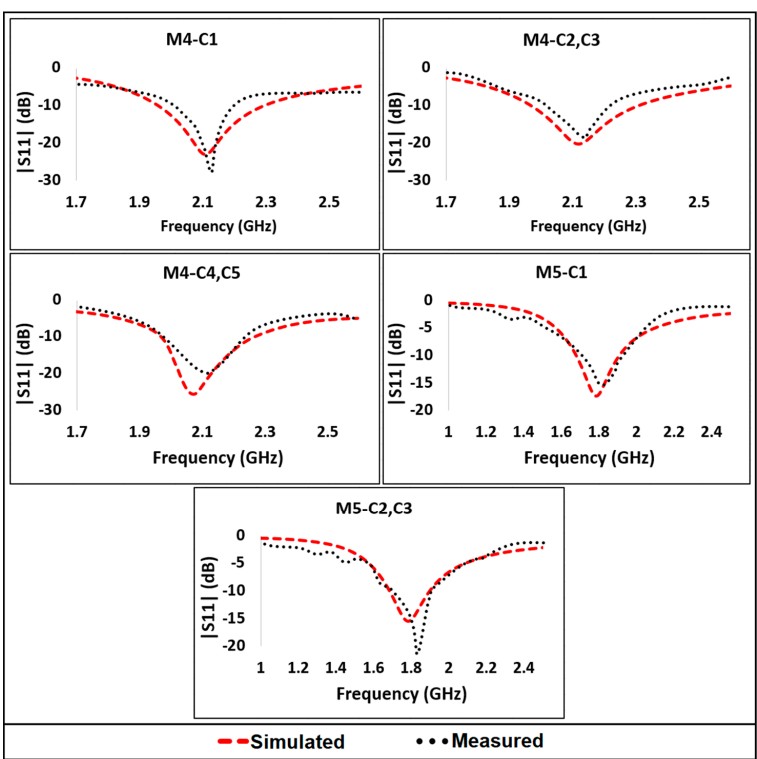

**Figure 9.** Measured vs. Simulated |S11| for Cases of Modes 4 and 5.

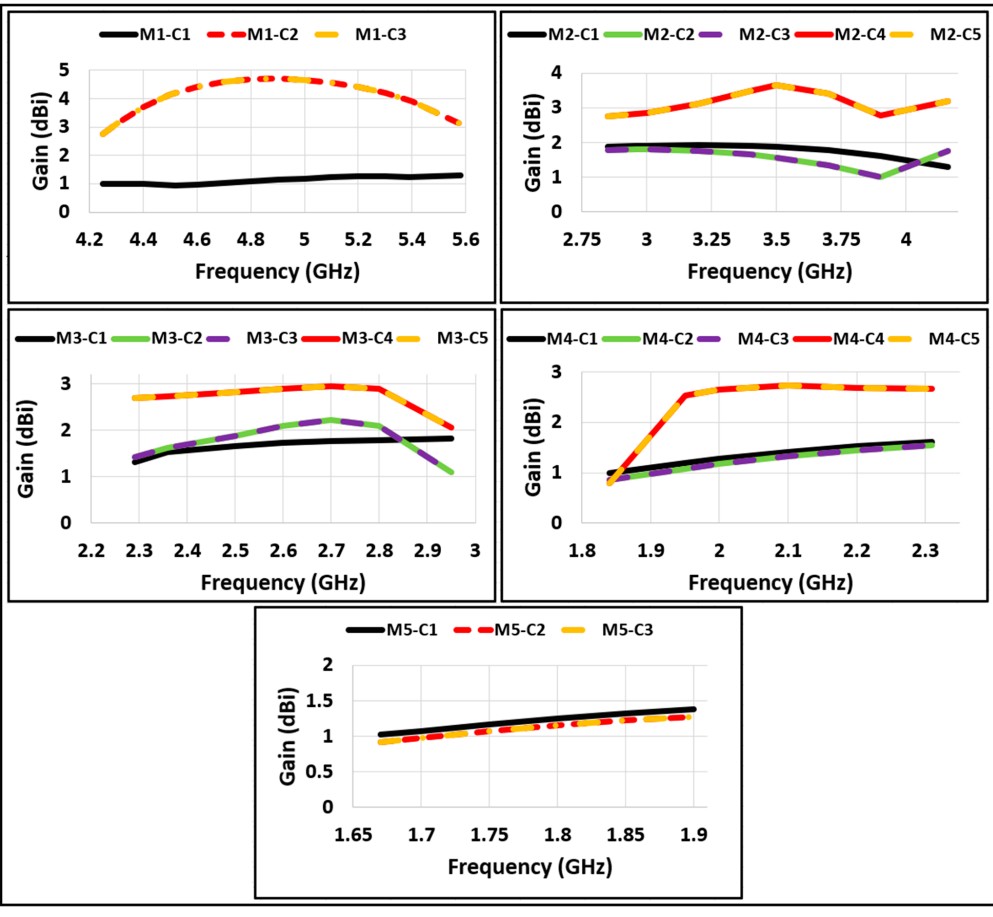

**Figure 10.** Gain vs. frequency plots for all Modes and Cases of Operation.

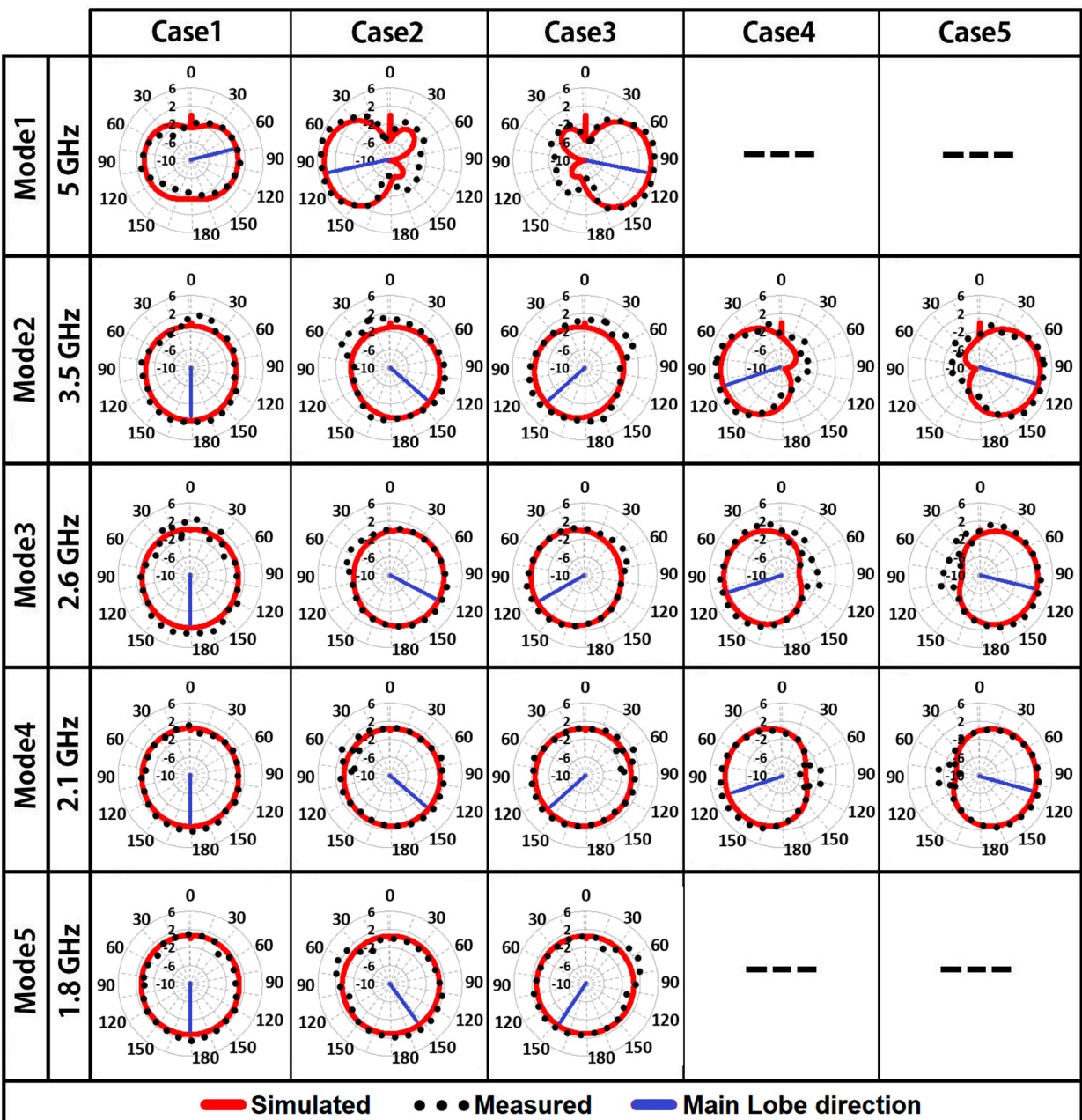

**Figure 11.** Polar plots for all modes (at Phi = 0, x–z plane).

### 3.1. Operation in Mode 1

When none of the switches are ON (i.e., S1−S12 = OFF), the antenna operates at 5 GHz Band (4.52–5.39 GHz). We term this as Case 1 of Mode 1. In this scenario, the antenna directs the main beam of the H-plane pattern (i.e., Phi = 0) at 85 °, as shown in Figure 11. Case 2 of Mode 1 is activated when switches S6 and S7 are ON, enabling the antenna to operate in a slightly wider frequency range (4.25–5.60 GHz). In this case, the main beam is steered along −95°. Case 3 of Mode 1 is in effect when S10 and S11 are ON. While the operating frequency range is the same as in Case 2, the beam is now steered along + 95°.

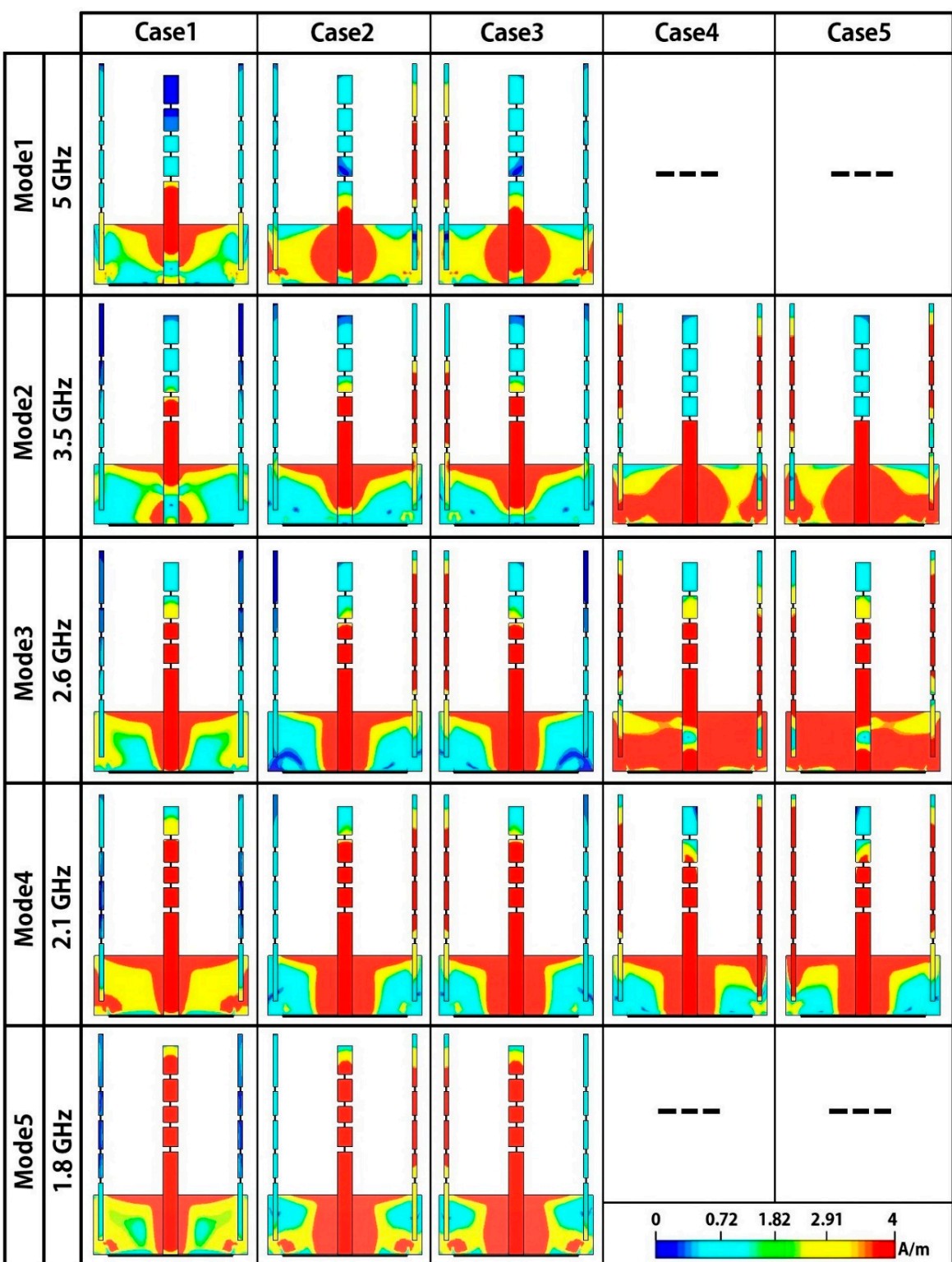

**Figure 12.** Surface currents distribution of all modes.

### 3.2. Operation in Mode 2

Case 1 of Mode 2 is enabled when switches S2–S4 are OFF but S1 is ON. The antenna now switches its operating frequency band to 3.5 GHz (2.96–4.17 GHz Range), directing the main beam at 180° (at Phi = 0). When switches S1, S6, and S7 are ON, Case 2 of Mode 2 is selected. Case 3 is activated when S1, S10, and S11 are ON. In both these cases, the antenna

operates in a comparatively smaller range of 3.00–3.92 GHz. However, for 3.5 GHz, the main beam of the antenna is directed along +140° and −140° for Cases 2 and 3, respectively. When switches S5–S8, S11, and S12 are ON (in addition to S1), Case 4 is selected while Case 5 is activated when S1 and S7–S12 are ON. The antenna operates within the 2.85–3.67 GHz range in these two Cases. For 3.5 GHz, the beam is directed along −100° in Case 4 and +100° in Case 5.

### 3.3. Operation in Mode 3

Case 1 of Mode 3 is enabled when the first two switches (i.e., S1 & S2) are ON. In this case, the operating frequency band is 2.6 GHz (2.36–2.95 GHz), while the beam is directed at 180°. When switches S6–S8 are switched ON, in addition to S1 and S2, the antenna starts operating in Case 2 of Mode 3. In this case, the operating range is slightly reduced (2.40–2.90 GHz), while the beam shifts to +117° for a frequency of 2.6 GHz. When switches S10–S12 are ON (along with S1 & S2), the 3rd Case of Mode 3 is activated. The operating range is the same as in Case 2, but for 2.6 GHz, the beam is steered along −117°. Turning ON Switches S1, S2, S5–S8, and S10–S12 selects Case 4 while turning ON switches S1, S2, and S6–S12 enables Case 5. The operating range of the antenna in both these cases is slightly reduced to 2.29–2.80 GHz, while for 2.6 GHz, the beam shifts to −103° and +103° for Cases 4 and 5, respectively.

### 3.4. Operation in Mode 4

When the main radiator switches S1, S2, and S3 are turned ON, Case 1 of Mode 4 is selected. The antenna operates at 2.1 GHz Band (1.95–2.30 GHz), while the beam is directed along 180°. The operation shifts to the Case 2 of Mode 4, when switches S6–S8 are ON, in addition to S1–S3. In this mode, the antenna operates within 1.96–2.31 GHz range (similar to case 1) but the beam is directed along +146° at 2.1 GHz. Turning ON switches S1–S3 and S10–S12, selects the 3rd Case of Mode 4. The operating frequency range is the same as Case 1 (1.95–2.30 GHz), while the beam is steered along −146°. When switches S5–S8, S10–S12 are ON, along with SS1–S3, Case 4 of Mode 4 is in effect, in which the operating frequency range is slightly reduced (1.97–2.28 GHz) and the beam is directed at −107°. The operation shifts to Case 5 of Mode 4 when switches S1–S3 and S6–S12 are ON. The operating frequency range is the same as Case 4 (1.97–2.28 GHz) but now the beam is steered along +107°.

### 3.5. Operation in Mode 5

When all the switches of the main radiator element, S1–S4, are ON, Case 1 of Mode 5 is enabled. At 1.8 GHz, the antenna directs the beam along 180°, while covering 1.8 GHz Band (1.67–1.90 GHz). Case 2 is selected when switches S6–S8 are ON, in addition to S1–S4, while Case 3 is selected when Switches S1–S4 and S10–S12 are ON. The antenna operates within the 1.68–1.90 GHz range in both Cases but steers the beam along +154° −154° in Cases 4 and 5, respectively.

The performance of the proposed antenna in different modes and cases is summarized in Table 3, which includes the operating frequency band, the specific frequency range within the band, peak gain in every sub-band, as well as the direction of the main beam. The "beam shifting angle" in Table 3 indicates the extent of beam shifting (in degrees), when compared to Case 1, which acts as a reference for every mode.

**Table 3.** Performance parameters of design structure.

| Mode | Case and Switch Configuration | Operating Frequency Band (GHz) | Frequency Range $f_l$–$f_u$ (Bandwidth in MHz) | Peak Gain (dBi) | Main Beam Direction (at $\varphi = 0°$) | Beam Steering Angle |
|---|---|---|---|---|---|---|
| 1 | (1) When all Switches are off | | 4.52–5.39 (870) | 1.28 | +85° | 0° |
| | (2) When S6 & 7 are on | 5 | 4.25–5.60 (1350) | 4.67 | −95° | −180° |
| | (3) When S10 & 11 are on | | 4.25–5.60 (1350) | 4.67 | +95° | +10° |
| 2 | (1) When only S1 is on | | 2.96–4.17 (1210) | 1.87 | 180° | 0° |
| | (2) When S1, 6 & 7 are on | | 3.00–3.92 (920) | 1.57 | +140° | +40° |
| | (3) When S1, 10 & 11 are on | 3.5 | 3.00–3.92 (920) | 1.57 | −140° | −40° |
| | (4) When S1, 5, 6, 7, 8, 11 & 12 are on | | 2.85–3.67 (820) | 3.64 | −100° | −80° |
| | (5) When S1, 7, 8, 9, 10, 11 & 12 are on | | 2.85–3.67 (820) | 3.64 | +100° | +80° |
| 3 | (1) When S1 & 2 are on | | 2.36–2.95 (590) | 1.74 | 180° | 0° |
| | (2) When S1, 2, 6, 7 & 8 are on | | 2.40–2.90 (500) | 2.11 | +117° | +63° |
| | (3) When S1, 2, 10, 11 & 12 are on | 2.6 | 2.40–2.90 (500) | 2.11 | −117° | −63° |
| | (4) When S1, 2, 5, 6, 7, 8, 10, 11 & 12 are on | | 2.29–2.80 (510) | 2.9 | −103° | −77° |
| | (5) When S1, 2, 6, 7, 8, 9, 10, 11 & 12 are on | | 2.29–2.80 (510) | 2.9 | +103° | +77° |
| 4 | (1) When S1, 2 & 3 are on | | 1.95–2.30 (350) | 1.51 | 180° | 0° |
| | (2) When S1, 2, 3, 6, 7 & 8 are on | | 1.96–2.31 (350) | 1.33 | +146° | +34° |
| | (3) When S1, 2, 3, 10, 11 & 12 are on | 2.1 | 1.96–2.31 (350) | 1.33 | −146° | −34° |
| | (4) When S1, 2, 3, 5, 6, 7, 8, 10, 11 & 12 are on | | 1.97–2.28 (310) | 2.80 | −107° | −73° |
| | (5) When S1, 2, 3, 6, 7, 8, 9, 10, 11 & 12 are on | | 1.97–2.28 (310) | 2.80 | +107 ° | +73° |
| 5 | (1) When S1, 2, 3 & 4 are on | | 1.67–1.90 (230) | 1.25 | 180° | 0° |
| | (2) When S1, 2, 3, 4, 6, 7 & 8 are on | 1.8 | 1.68–1.90 (220) | 1.15 | +154° | +26° |
| | (3) When S1, 2, 3, 4, 10, 11 & 12 are on | | 1.68–1.90 (220) | 1.15 | −154° | −26° |

An analysis of the return loss (S11) plots shows that the matching impedance of the antenna is good for all desired frequency bands, in all modes and cases. Figures 7–9 compare simulated and measured return losses, while Figure 10 shows gain vs. frequency plots for all cases of respective modes. The proposed antenna's gain ranges from 1.15 to 4.67 dBi with peak values of 1.25, 2.80, 2.9, 3.64 and 4.67 dBi over respective operating frequencies. Measured and simulated results are in good agreement with each other. The variations between simulated and measured results are observed due to fabrication and measurement tolerances. The measurements were conducted at the antenna measurement laboratory of the National University of Science and Technology (NUST) Islamabad.

The surface currents distributions for all modes are depicted in Figure 12. These plots provide insight into the operation of the antenna in various operating modes and cases, indicating resonant areas of the antenna structure contributing to radiation. The surface currents distributions of Case 1 of all operating modes, indicated by the first column of Figure 12, are related to frequency reconfigurability. The plots also indicate the contributing resonant lengths, for respective operating frequencies. The Case 1 in every mode, which acts as a reference for that mode, indicates that the operating frequency range decreases as the resonant length for the respective frequency band increases, thus supporting the inverse relation between resonant length and operating frequency. All other current distribution plots inform about the contributing lengths of parasitic elements for

different pattern reconfiguration cases. Cases 2 to 5 of Mode 2, 3 and 4 and Cases 2 and 3 of Modes 1 and 5 show the active portions of designed parasitic elements at different cases and indicate their behavior (reflector, director or combination of both) that results in beam shifting accordingly at those cases. Analyzing the Surface currents distributions concurrently with polar plots provides a clearer understanding of pattern reconfiguration operation in each case. In order to fully comprehend the reconfigurability operation, Mode 1 and Mode 2 are considered. In Mode 1–Case 1, when no switch is ON, the antenna is operating at 5 GHz and radiating in the direction of 85° (at a cut angle of Phi = 0°) and none of the portions of parasitic elements contributes to the operation, which can be seen in its respective surface currents distribution. In Mode 1–Case 2, when switches S6 and S7 are ON, the respective surface currents distribution indicates that the right-side electromagnetically coupled parasitic element act as a reflector for 5GHz frequency and shifts the beam in −95°. In Mode 1–Case 3, when switches S10 and S11 are ON, the surface currents distribution shows that the left-side electromagnetically coupled parasitic element acts as a reflector and shifts the pattern in +95°. In Mode 2–Case 1, when switch S1 is on only, the antenna is radiating in 180° at 3.5GHz and none of the section of parasitic elements is in action as shown by its respective surface currents distribution. In Case 2 of this mode, when switches S6 and S7 are ON, the surface current plot indicates that the right-side electromagnetically coupled parasitic elements act as director for 3.5 GHz and direct the pattern at +140°. In Case 3, when S10 and S11 are ON, the surface currents show the left-side electromagnetically coupled parasitic elements act as a director and shift the pattern in −140° for 3.5 GHz. In Mode 2-Case 4, when S1, S5, S6, S7, S8, S11 and S12 are ON, the respective surface currents distribution plot indicates that the left-side director and right-side reflector come into action collectively and shifts the beam in −100°. In Case 5, when S1, S7, S8, S9, S10, S11and S12 are ON, the right-side director and left-side reflector come into action as indicated by respective surface currents, and the antenna shifts the beam in +100° for 3.5 GHz. Similarly, the surface currents distribution plots of all cases of the other three modes (Mode 3,4 and 5) clearly explain the complete reconfigurability operation of the antenna at those cases. Table 3 summarizes the overall performance of the antenna in terms of various performance parameters and the performance metrices of the proposed antenna are compared with the latest reported work in Table 4. The comparison table demonstrates that this antenna is relatively compact in size, with greater number of working bands, better gain, and better bandwidth.

**Table 4.** Comparison of the performance of the proposed antenna with other relevant designs.

| Ref. No. | Size (Ls × Ws) | No. of switches | Type of Switches | No. of Operating Frequency Bands | Max Number of Beams | Bandwidth (MHz) | Peak Gains (dBi) |
|---|---|---|---|---|---|---|---|
| [32] | 45 × 50 | 2 | Pin diode | 1 | 3 | 800 | 2.2 |
| [33] | 160.9 × 151.5 | 2 | Varactor | 2 | 2 | 730 | 9 |
| [34] | 112 × 52 | 18 | NMOs | 2 | 8 | 1400/1501 | 3.8/8.3 |
| [35] | 113 × 113 | 14 | Pin diode | 2 | 3 | N/A | 6.2/6.6 |
| [36] | 23 × 31 | 3 | Pin diode | 2 | 4 | 1700/1200 | 4.01/4.60 |
| [37] | 42 × 44 | 8 | Pin diode | 2 | 2 | 160/220 | NG |
| [38] | 40 × 30 | 4 | Pin diode | 2 | 3 | 400/500 | 2.24/2.76 |
| This work | 46 × 32 | 12 | Pin diode | 5 | 5 | 230/350/590 /1210/1350 | 1.25/2.80/2.90 /3.64/4.67 |

NG = Not Given.

## 4. Conclusions

This work has proposed, designed, and evaluated the successful operation and performance of a penta-band antenna offering both frequency and pattern reconfigurability. Twelve pin-diodes switches are used to achieve compound reconfigurability; four switches (S1 to S4) enable frequency reconfigurability, while eight other switches (S5 to S12) allow and control beam steering. There are five modes of operation of this antenna. Each mode

has several cases, offering a diverse range of beam steering options over the operating frequency band. The antenna operates at 5 GHz Band (4.52–5.39 GHz) in Mode1 (when all switches are OFF). The operating band switches to 3.5 GHz and 2.6 GHz, in Mode 2 (when S1 is ON only) and Mode 3 (when S1 and S2 are ON), respectively. Similarly, in Mode 3 (when S1 to S3 are ON) and Mode 4 (when S1 to S4 are ON) operation shifts to 2.1 and 1.8 GHz, respectively. The antenna offers multi-band operation in several sub-6 GHz 5G bands (2.1, 2.6, 3.5, and 5 GHz). It also supports GSM (1.8 GHz), UMTS (2.1 GHz), 4G-LTE (2.1 GHz and 2.6 GHz), WiMAX (2.6 GHz and 3.5 GHz), and WLAN (5 GHz) applications.

**Author Contributions:** Conceptualization, S.U. (Sadiq Ullah) and W.U.R.K.; methodology, I.A., T.A., H.D. and S.U. (Sadiq Ullah); software, H.D.; validation, I.A. and S.U. (Shakir Ullah); formal analysis, B.K.; investigation, S.U. (Sadiq Ullah), A.G., M.I.H.; resources, W.U.R.K., S.U. (Sadiq Ullah), and T.A.; writing—original draft preparation, I.A.; writing—review and editing, I.A.; visualization, H.D. and N.M.; supervision, W.U.R.K.; project administration, S.U.; funding acquisition, M.I.H. and A.G. All authors have read and agreed to the published version of the manuscript.

**Funding:** This research received no external funding.

**Data Availability Statement:** Data is contained within the article.

**Conflicts of Interest:** The authors declare no conflict of interest.

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
