# Peer review of "A Pentaband Compound Reconfigurable Antenna for 5G and Multi-Standard Sub-6GHz Wireless Applications"

_electronics, doi:10.3390/electronics10202526_

Round 1

Reviewer 1 Report

In this article, the authors report reconfigurable and beamsteered antenna printed on FR4 substrate. The frequency and pattern reconfigurability was enabled through PIN diode switches. The manuscript can be further strengthened by addressing the following issues:

(a) The authors need to explain the term 'low profile'.
(b) The authors need to clear about the different FCC band spectrum
(c) The reference to other literature are not properly cited. Instead of reporting 'In [12]', the investigators can mention the lead authors name in the text.
(d) Recent articles, especially, review articles on reconfigurable antennas have not been cited.
(e) The biasing circuit shown in figure 3 lacks clarity and seems to have electrical shorts.
(f) The zoomed in image of the pin diode is necessary.
(g) Figure 4 showing the anechoic chamber seems dubious.
(h) The variations in the simulated vs measured return loss in figure 5 has not been explained.
(i) Figure 8 needs explanation. 
(j) The comparison table (table 4) indicates much smaller reconfigurable antennas have been realized earlier. 

Author Response

Attached kindly find our reply

Reviewer 2 Report

My comments are listed below: 1. Introduction : The author only summarized the features of prior works in this section. The authors ware supposed to highlight the motivation and originality of your work compared to highly related works [30-37]. I recommend to add the limitations of those works and explain what problem the authors want to address. 2. Fig. 1 : please include the xyz-coordinate in Fig. 1. 3. For in-depth understanding of the impact of design parameters on antenna performances, the authors need to add parametric studies on some important design parameters (e.g., L11, line width, etc) 4. Sec. III : In mode 1, instead of other modes, the main beam is formed at nearly 90 [deg] for case #1. Could you explain why it happens? Is it acceptable in terms of pattern reconfigurability? 5. Sec. III : In mode 5, it seems that the front-back-ratio gets reduced for cases #2 and #3. Could you explain this in detail?

Author Response

Attached kindly find our reply

Reviewer 3 Report

Thank you for submitting your work on pentaband reconfigurable antenna. 

I have a few suggestions:

  1. Please add an axis system ( XYZ mini axis) in Figure 1 so that the radiation pattern plots could be related for XZ planes in Figure 9
  2. Please add the following paper in reference: Hasan MN, Seo M. Compact omnidirectional 28 GHz 2× 2 MIMO antenna array for 5G communications. In2018 International Symposium on Antennas and Propagation (ISAP) 2018 Oct 23 (pp. 1-2). IEEE.
  3.  

Author Response

Attached kindly find our reply

Reviewer 4 Report

Authors have presented manuscript titled “A Pentaband Compound Reconfigurable Antenna for 5G and 2 Multi-Standard Sub-6GHz Wireless Applications”. The manuscript is well presented. Following will be helpful to further improve the manuscript.

  • Replace “Returnloss (dB)” with “|S11| (dB)” in figures 5, 6, 7. Also update the captions and text accordingly.
  • Use same scale for the radiation patterns (optional, for better understanding) in figure 9.
  • Add a column in Table 4 for type of switching device used (i.e. PIN etc) in case they are using different switching devices.

Author Response

Attached kindly find our reply

Round 2

Reviewer 1 Report

The response to comment 1 should be included in the manuscript.

The authors need to explain the High band, Mid-band and low-band of the 5G spectrum.

It is good practice to mention the lead author's name when you reference literature. I recommend authors to change the way they reference literature throughout the manuscript.

Author Response

Comment 1: The response to comment 1 should be included in the manuscript. “The authors need to explain the term 'low profile'”.

Response 1: We are thankful to the reviewer for the comment. The response to comment 1 has been added in the manuscript.

Comment 2: The authors need to explain the High band, Mid-band and low band of the 5G spectrum.

Response 2:  We are thankful to the reviewer for mentioning this. The Low-band spectrum is any spectrum that is lower than 1 GHz on the spectrum chart. Spectrum in the 1 GHz - 6 GHz range is mid-band spectrum. The high band on the spectrum chart in the 24 GHz band and higher.

Comment 3: It is good practice to mention the lead author's name when you reference literature. I recommend authors to change the way they reference literature throughout the manuscript.

Response 3: We are thankful to the reviewer for this suggestion. This is actually the nice suggestion, but we are following the MDPI journal referencing style, and also did not find lead author name in the literature referencing style in MDPI journals papers.

Reviewer 2 Report

The revision were made sufficiently. All my comments were reflected in the revised manuscript. 

Author Response

Comment 1: The revision were made sufficiently. All my comments were reflected in the revised manuscript. 

Response 1: We are thankful to the reviewer for positive response.